# Factors associated with phylogenetic clustering of hepatitis C virus, mainly among people who inject drugs who access HIV prevention services in South Africa, 2016–2017

Nkosenhle L. Ndlovu[1,2¤a]*, Andrew Scheibe[3,4], Harry Hausler[3], Mark W. Sonderup[5], C. Wendy Spearman[5], Katherine Young[3], Dawie Nel[6], Jason T. Blackard[7☯], Nishi Prabdial-Sing[1,2☯¤a]

1 School of Pathology, Faculty of Health Sciences, University of the Witwatersrand, Johannesburg, South Africa, 2 National Institute for Communicable Diseases, Centre for Vaccines, Johannesburg, South Africa, 3 TB HIV Care, Cape Town and Durban, South Africa, 4 Community Oriented Primary Care Research Unit, Department of Family Medicine, University of Pretoria, Pretoria, South Africa, 5 Division of Hepatology, Department of Medicine, Faculty of Health Sciences, University of Cape Town, Cape Town, South Africa, 6 OUT LGBT Well-being, Pretoria, South Africa, 7 University of Cincinnati College of Medicine, Cincinnati, Ohio, United States of America

☯ These authors contributed equally to this work
¤a Current Address: National Institute for Communicable Diseases, Centre for Vaccines, 1 Modderfontein Rd, Sandringham, Johannesburg, 2192, South Africa
* nkosenhlen@nicd.ac.za

## Abstract

People who inject drugs (PWID) are disproportionately burdened with hepatitis C virus (HCV) infection in South Africa (SA). Transmission dynamics can be inferred using phylogenetic clustering to inform prevention interventions. We utilized Core-E2 sequences and demographic data to investigate factors associated with HCV phylogenetic clustering among PWID and men who have sex with men (MSM) who inject drugs in SA. Previously genotyped samples (n = 285) that met the selection criteria were extracted, amplified, and Sanger sequenced. Phylogenetic trees were inferred using maximum likelihood implemented in RAxML (Cipres Gateway). Transmission clusters were determined in Clusterpicker using a 90% bootstrap threshold and a genetic distance cut-off on genetic similarity of ≤3.5%. Factors associated with clustering were assessed using logistic regression. Phylogenetic clustering of Core-E2 sequences was observed for 55% (78 of 141) of participant samples that were successfully sequenced, including 50 (64.1%) with genotype 1a and 28 (35.9%) with genotype 3a. Twelve clusters were identified, including six clusters each for genotypes 1a and 3a. Among genotype 1a, the cluster size ranged from 3 to 15 participants. Among genotype 3a, the cluster size ranged from 3 to 9 participants. Clustering among the mixed ancestry group in Cape Town was noted for ages 18–55. Factors independently associated with phylogenetic clustering included sharing a needle (adjusted odds ratio [aOR] 4.02, 95% confidence interval [CI] 1.08–14.87,

**Data availability statement:** The minimized metadata associated with the sequences reported in this paper have been deposited in the Open Science Framework (OSF) repository and are publicly accessible at DOI: 10.17605/OSF.IO/J82M9. The sequence data, corresponding to accession numbers PX377912 through PX378052, have been submitted to the National Centre for Biotechnology Information (NCBI) database.

**Funding:** NLN was awarded the National Research Foundation (DST-NRF Innovation) student bursary (SFH180614346729). NLN was awarded the Poliomyelitis Research Foundation (PRF) student bursary (18/18). NPS was awarded the National Health Laboratory Service Research Trust (NHLSRT) grant (2018-1DEV46-NPR03). AS received funding from the Bristol-Myers Squibb Foundation. For more information, see: National Research Foundation: https://www.nrf.ac.za/ Poliomyelitis Research Foundation: https://prf.ac.za/ National Health Laboratory Service Research Trust: https://www.nhls.ac.za/services/research/nhls-research-trust/ Bristol-Myers Squibb Foundation: https://www.bms.com/about-us/responsibility/bristol-myers-squibb-foundation.html The funders were not involved in the design of the study, nor in the collection, analysis, or interpretation of data.

**Competing interests:** The authors have declared that no competing interests exist.

p = 0.037), age ≥ 29 years (aOR 3.00, 95% CI 1.22–7.37, p = 0.016), and mixed ancestry race (aOR 6.11, 95% CI 1.87–19.95, p = 0.003). These data highlight the urgent need to reduce transmission by providing sufficient sterile needles and syringes and tailored education to prevent HCV transmission among older, experienced PWID.

## 1. Introduction

The hepatitis C virus (HCV) infection is a major global health problem that has affected approximately 1% of the human population [1]. A 2022 meta-analysis reported 56.8 million viremic infections globally [2]. HCV is a leading cause of hepatocellular carcinoma (HCC) and liver transplantation [1]. Globally, 0.98 million (0.76–1.34 million) new chronic cases and 244,000 (197,000–288,000) deaths were reported in 2022 [3]. Fifteen to twenty-five percent of infections clear during the acute phase, while 75–85% progress to chronic infection [4]. Due to the frequent asymptomatic nature of infection, most infections go undiagnosed [5,6]. With the effect of limited access to screening resources, limited awareness, and a lack of active surveillance systems, only 36.4% of infected individuals globally knew their HCV status in 2022 [5].

HCV is a member of the *Flaviviridae* family and genus *Hepacivirus* [7]. The open reading frame (ORF) is 9030–9099 nucleotides in length depending on the HCV genotype and encodes a polyprotein of 3010–3033 amino acids [8]. The proteins encoded by the ORF are grouped into structural and non-structural proteins. The structural proteins – including Core (C), Envelope 1 (E1), and Envelope 2 (E2) – are involved in protecting the viral ribonucleic acid (RNA) and viral entry [8]. The non-structural proteins – NS1, NS2, NS3, NS4A, NS4B, NS5A, and NS5B – are responsible for polyprotein modification, RNA replication, and viral assembly [8].

HCV is a positive-sense virus classified into eight genotypes based on nucleotide sequence diversity [7]. HCV genotypes 1, 3, and 4 are the most common worldwide, accounting for 44%, 25%, and 15% of all infections, respectively [1]. Genotypes 2, 6, and 5 account for 13%, 2%, and 1% of all infections globally [1]. Genotypes 1 and 3 are widely distributed except for Central Sub-Saharan Africa. South Africa has a pan-genotypic distribution with genotypes 1, 2, 3, 4, 5, and mixed genotypes accounting for 31.5%, 1.2%, 12.6%, 12.4%, 35.7%, and 6.7% of infections, respectively [1].

HCV disproportionally affects key populations, including PWID and MSM [9–12]. PWID are exposed to the infection by sharing injecting needles and preparation equipment [9]. MSM are at risk of infection due to high-risk sexual practices [9]. Previous studies have supported intermixed risk factors such as injecting drug use and risky sexual practices within networks of HIV-infected MSM [13].

In SA, as observed globally, HCV predominates in key population groups, notably HIV-positive MSM and PWID. In a 2017 study conducted in MSM who inject drugs in Cape Town, the anti-HCV seroprevalence was 27% (11 of 41) [10]. Similarly, a 2019 study in male PWIDs in Pretoria demonstrated a viremic rate of 83% (196 of 237) [11]. In contrast, a systematic review and meta-analysis showed an anti-HCV

seroprevalence < 1% in the general population [14]. Additionally, from a cross-sectional study on key populations, the genotype distribution on PWID was restricted to genotypes 1a and 3a [11] compared to 5a and 1 among patients with chronic liver disease and blood donors [15].

While conventional epidemiological studies have been invaluable for investigating factors associated with disease acquisition [11], they cannot fully examine community transmission dynamics, especially for diseases with long incubation periods and high rates of asymptomatic infection, such as HCV [16,17]. Molecular epidemiology combines genetic sequencing techniques with traditional epidemiological methods and enables genetic linking of related cases when contact-tracing information is unavailable or unreliable [18]. This approach has been successfully applied to HIV and HCV studies, where phylogenetic clustering data has been used to map transmission networks [16,18,19]. By integrating molecular techniques with traditional epidemiologic approaches, researchers can construct more accurate transmission models, identify outbreak sources, and track pathogen evolution over time. This interdisciplinary approach provides a powerful tool for understanding complex infectious disease dynamics and informing targeted public health interventions.

The study employed phylogenetic clustering analysis to investigate HCV transmission networks and associated factors among PWID in SA. This molecular epidemiological approach examined factors linked to phylogenetic clustering within these populations. In the context of SA, where drug use remains stigmatized and criminalized, the molecular characterization of HCV among PWID has the potential to inform public health preventive strategies. These strategies may include infection control measures, awareness campaigns, education about high-risk behaviors, the provision of needle and syringe programs (NSP), and opioid substitution therapy (OST) to those most in need. The findings from this study could contribute to evidence-based interventions tailored to the specific needs of PWID, potentially improving HCV prevention and treatment efforts in SA.

## 2. Methods

### 2.1. Study population and design

Samples and data used in this study were from a larger cross-sectional study examining viral infections in PWID and MSM in SA [10–12]. PWUD/ID (people who use drugs including people who inject drugs) were recruited from cities that operated harm reduction services at a PWUD (people who use drugs):PWID (people who inject drugs) ratio of 1:4 due to the increased risk of blood borne infection transmission through injecting practices [11]. The majority of samples came from PWID; however, the PWUD may have included people with previous experiences of injecting, although this was not explicitly assessed. HCV genotyping, HIV rapid, and HBV rapid test results were described previously [10,11]. Samples with an HCV viral load <1,000 IU, insufficient volume (<140 μL), and/or missing clinical data were excluded from the current study. Among PWID, minor genotypes such as 1b and 3c and mixed genotypes, as well as individuals with no genotype result were excluded. Samples for MSM were collected from two sources – patients attending a specialist clinic in Cape Town and participants in a cross-sectional study conducted in Johannesburg. The archived samples were accessed for research purposes on 21 January 2021. The authors did not have access to information that could identify individual participants during or after data collection. This study was approved by the University of the Witwatersrand Human Research Ethics Committee (clearance certificate M181169).

### 2.2. HCV RNA extraction and cDNA synthesis

RNA was extracted from 140 μL of EDTA plasma using the QIAmp Viral RNA Mini kit (Qiagen Pty Ltd, Hilden, Germany). According to the manufacturer's instructions, RNA was eluted in 60 μL of AVE buffer. Reverse transcription for the production of complementary DNA (cDNA) was conducted as described previously [20]. The master mix composition for reverse transcription with minor modifications included: SuperScript VILO™ with random hexamers 4.0 μL (Cat No.11754250, Invitrogen, California, USA), Superscript enzyme mix 2.0 μL (Invitrogen, California, USA), nuclease-free water 4.0 μL, and

RNA 10.0 µL. Cycling conditions were: preheated at 25°C for 10 min, reverse transcription at 42°C for 1 hour, and enzyme deactivation at 85°C for 5 min.

### 2.3. Core-E2 PCR amplification

The Core-E2 PCR amplification was conducted using a previously described method [20]. Primers targeted the 5' UTR–HVRI region of the HCV virus generating a 1.5 kbp fragment [21]. The master mix composition for PCR 1 included nuclease-free water 12.8 µL, HI-FI buffer (Bioline Pty Ltd, Alexandria, NSW Australia) 4.0 µL, dNTP 0.2 µL, velocity polymerase enzyme (Bioline Pty Ltd, Alexandria, NSW Australia) 0.4 µL, primer HCVuniv134S22 0.2 µL, primer HCVuniv1987A22 0.4 µL, and cDNA 2.0 µL.

PCR 1 cycling conditions were preheated at 98°C for 2 min, denaturing at 98°C for 30 s, annealing at 60°C for 30 s among genotype 3a samples, annealing at 56°C for 30 s among genotype 1a samples, extension at 72°C for 1 min, and final extension at 72°C for 10 min. PCR 2 was conducted for all samples. PCR 2 master mix contained nuclease-free water 25.6 µL, HI-FI buffer 8.0 µL, dNTP 0.4 µL, velocity polymerase enzyme 0.8 µL, primer HCVuniv278S22 0.4 µL, primer HCVuniv1791A22 0.8 µL, first round PCR product 4.0 µL. PCR 2 cycling conditions were preheated at 98°C for 2 min, denaturing at 98°C for 30 s, annealing at 56°C for 30 s, extension at 72°C for 1 min, and final extension at 72°C for 10 min.

### 2.4. Gel electrophoresis and PCR purification

Amplified products were confirmed by gel electrophoresis (Promega, Corporation, Madison, WI USA) against molecular marker VI (0.15–2.1Kbp, Roche, Mannheim, Germany) under a UV transilluminator and Genesnap (Syngene, Cambridge, United Kingdom). Twenty-two µL of the amplicon was loaded with 2.5 µL of Gel Pilot Loading Dye 5X (Qiagen, Hilden, Germany). PCR positive amplicons were visualized on 1% agarose gel and extracted/cleaned using a QIAquick® Gel Extraction Kit (Qiagen, Hilden, Germany).

### 2.5. Sanger sequencing

According to the manufacturer's instructions, Sanger sequencing was conducted using BigDyeTM Terminator 3.1 kit (ThermoFisher, California, USA). The master mix had a total of 10 µL, and PCR 2 amplification primers were used as sequencing primers. Further, the PCR master included sequencing primer 1.0 µL, BigDye Terminator 4.0 µL, Purified PCR 2 amplicon 1.0 µL, and nuclease-free water 4.0 µL. Sequencing reactions were purified using BigDye XTerminator ™ (ThermoFisher, California, USA) according to the manufacturer's instructions. Sequencing was conducted on the ABI 3500 (Thermofisher, USA). Sequence chromatograms were assessed in Sequencher (Gene Codes Corporation, Ann Arbor, USA).

### 2.6. Phylogenetics

After editing the HVR1 and ambiguous nucleotides, the sequences analyzed were 1104 bases in length. The HVR1 was removed due to the high genetic variability of this region, which interferes with pair and cluster identification [22]. Study sequences and the Smith panel of reference sequences were aligned using Mafft [23]. The codon Align tool [24] was used to put sequences in the correct reading frame, and the alignment was checked manually using BioEdit [25].

Phylogenetic trees of the Core-E2 without HVR1 were inferred together with study sequences and the Smith panel of reference sequences using maximum-likelihood analysis implemented in RAxML through CIPRES Science Gateway [26]. Reference sequences were included to support local cluster identification [18,22]. Using JModelTest [27] to determine the most appropriate model of nucleotide substitution, the General Time Reversible model of nucleotide substitution with a gamma-shaped distribution rate variation across sites was selected for inferring phylogenetic trees (GTR + G) with 1,000

replicates. To select a genetic distance threshold, a sensitivity analysis was conducted with genetic distances of 1.5%, 3.5%, and 5%. ClusterPicker [28] was used to identify clusters with bootstrap values ≥90% and a genetic distance threshold ≤3.5%. Phylogenetic trees were annotated using the online tool Evolview [29] and visualized using Figtree [30].

### 2.7. Study outcome

Phylogenetic clustering of HCV infections was defined by clustering three or more HCV genomes of study participants with a bootstrap value ≥ 90% and a genetic distance threshold ≤3.5%.

### 2.8. Statistical analysis

Data analysis was performed on STATA 17 (Stata corp LLC, Texas, USA). The datasets for PWID and MSM were integrated using unique identifiers, ensuring precise linkage of laboratory results, demographic information, and behavioral data with corresponding sample information. Descriptive statistics were then employed to analyze the combined dataset, with results stratified by city, population, and phylogenetic clustering. Factors included in the logistic regression models were pre-determined from previous studies, either associated with HCV infection or phylogenetic clustering. Factors included residing in Pretoria, living on the street, injecting ≥ 4 times a day, HIV infection, black race, and being sexually active in the last month [11]. Other factors identified from studies in different countries were considered, including age [18], recent seroconversion [18], HIV infection [13,18], syringe borrowing [18], and HCV genotype 1a [13]. Variables with a $P$ value < 0.20 in the univariate analysis were included in the multivariate logistic regression analysis [13]. The backward stepwise approach was used to eliminate variables on the adjusted logistic regression model sequentially. Variables with $P$ value < 0.05 were considered statistically significant. Variables with a small number of observations were combined to produce new categories or were omitted from the logistic regression analysis.

## 3. Results

### 3.1. Participant characteristics

Among 411 participant samples with a viral load ≥1,000 IU, 285 samples were included in the current study (Fig 1). Each sample was tested twice following amplification or sequencing failure, and 184 of 285 (65%) were successfully amplified with the Core-E2 PCR, of which 141 (77%) were sequenced.

The descriptive characteristics of participants whose samples were sequenced in the current study were indicated from the larger study [10–12]. A summary of the participant characteristics of those samples sequenced in the Core-E2 fragment (n = 141) showed that PWID was the most dominant group (89%; n = 125/141, (Table 1). The age distribution showed a median of 31 years (range: 26–37), a mean of 41 (±7.4), and a mean of 30 (±5.8) among PWID, MSM, and PWUD, respectively. In our sampling, the HIV prevalence appeared to be highest among MSM (55%; n = 6/11), followed by PWUD (40%; n = 2/5) and PWID (28%; n = 35/125), with regional variations observed: HIV prevalence among PWID was 29% (n = 10/35) in Durban, 69% (n = 24/35) in Pretoria, and 3% (n = 1/35) in Cape Town, HBV infection rates were relatively low overall, at 6% (n = 8/125) among PWID, and 9% (n = 1/11) among MSM, with regional differences showing higher rates among PWID in Durban (9%; n = 3/34) compared to Pretoria (4%; n = 2/56).

From the samples that were successfully sequenced, the majority of participants were male (92.2%), though male representation was slightly lower in Pretoria (89.8%) compared to Durban (91.4%) and Cape Town (95.7%, Table 2). Racial composition varied regionally, with Black participants comprising 32.6% overall but concentrated in Pretoria (57.6%). Homelessness was prevalent among participants whose samples were sequenced (61%), particularly in Durban (71.4%). HCV genotype analysis revealed that genotype 1a was the most common (63.1%), followed by genotype 3a (34.8%) and genotype 4d (2.1%) exclusively found among MSM, with a consistent median HCV viral load of log 6 (range: 3–8) across locations (Table 2).

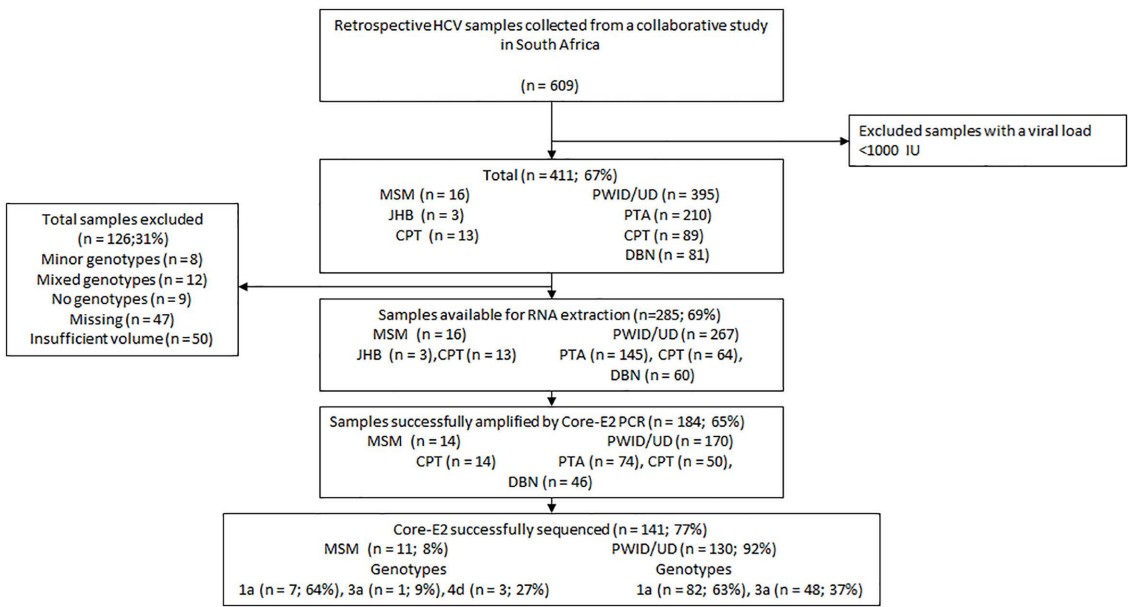

**Fig 1. Participant workflow.** PD represents PWID/UD, MM = men who have sex with men, CPT = Cape Town, DBN = Durban, and PTA = Pretoria.

**Table 1. Characteristics of participants with Core-E2 sequences.**

| Characteristic Total n (%) | PWID (n = 125) | MSM (n = 11) | PWUD (n = 5) |
|---|---|---|---|
| **Age (median, Q1–Q3)** | | | |
| Overall | 31 (26–37) | 41 (±7.4) | 30 (±5.8) |
| Durban | 29 (24–41) | N/A | N/A |
| Pretoria | 30 (26–36) | N/A | 33 (±4.1) |
| Cape Town | 33 (±7.4) | 41 (±7.4) | N/A |
| **HIV Infection n (%)** | | | |
| Overall | 35 (28.0%) | 6 (54.5%) | 2 (40.0%) |
| Durban | 10 (29.4%) | 0 (0.0%) | 1 (100.0%) |
| Pretoria | 24 (42.7%) | 0 (0.0%) | 1 (33.3%) |
| Cape Town | 1 (2.9%) | 6 (54.5%) | N/A |
| **HBV Infection n (%)** | | | |
| Overall | 8 (6.4%) | 1 (9.1%) | 1 (20.0%) |
| Durban | 3 (8.8%) | 0 (0.0%) | N/A |
| Pretoria | 2 (3.6%) | N/A | 1 (33.3%) |

"N/A" indicates no data.

Participants' substance use and sex practices were summarized in S1 Table **(with information regarding the larger study [9–11])**. Heroin was the most commonly used drug at 68.8% as compared to methamphetamine (28.4%). Heroin use was relatively high in all three cities, whilst methamphetamine use highest in Cape Town (36 of 40). In addition to 74.5% heroin use, 76.6% of participants from Cape Town also used methamphetamine. Of 141 participants, only 13.5% reported sharing a needle at the last injection (19 of 141). Fifty-five percent (78 of 141) of participants reported injecting

**Table 2. Characteristics of participants with Core-E2 sequences.**

| Characteristic Total n (%) | Overall (n = 141) | Durban (n = 35) | Pretoria (n = 59) | Cape Town (n = 47) |
|---|---|---|---|---|
| Log HCV viral load (median, Q1-Q3) | 6 (3–8) | 6 (5–6) | 6 (5–6) | 6 (5–6) |
| Male gender | 130 (92.2) | 32 (91.4) | 53 (89.8) | 45 (95.7) |
| PWID | 125 (88.7) | 34 (97.1) | 56 (94.9) | 35 (74.5) |
| PWUD | 5 (3.5) | 1 (2.9) | 3 (5.1) | 1 (2.1) |
| MSM | 11 (7.8) | 0 | 0 | 11 (23.4) |
| Race [n (%)] | | | | |
| Black | 46 (32.6) | 12 (34.3) | 34 (57.6) | 0 |
| Mixed ancestry | 32 (22.7) | 1 (2.9) | 2 (3.4) | 29 (61.7) |
| White | 59 (41.8) | 18 (51.4) | 23 (39.0) | 18 (38.3) |
| Indian | 4 (2.8) | 4 (11.4) | 0 | 0 |
| HCV genotype [n (%)] | | | | |
| 1a | 89 (63.1) | 20 (57.1) | 39 (66.1) | 30 (63.8) |
| 3a | 49 (34.8) | 15 (42.9) | 20 (33.9) | 14 (29.7) |
| 4d | 3 (2.1) | 0 | 0 | 3 (6.4) |
| Homeless (vs shelter/flat) | 86 (61.0) | 25 (71.4) | 34(57.6) | 27 (57.4) |

more than four times per day, and this number was comparable across the three cities. Eighty percent of participants reported injecting for a year.

### 3.2. Phylogenetic cluster composition

The overall phylogenetic tree includes HCV genotypes 1a (n = 90), 3a (n = 48), and 4d (n = 3) as shown in (Fig 2a and Fig 2b). Study sequences grouped with their respective genotypic counterparts from the Smith panel of reference sequences in confirmation of HCV genotype [7]. Smith's panel of references was included to support local cluster identification. Due to the high sensitivity cut-off on genetic similarity ≤3.5%, South African sequences from this study clustered separately from other references of the same genotype.

Fifty-five percent (78 of 141) of participants were involved in phylogenetic clusters (bootstrap threshold of ≥90% and a genetic distance threshold of ≤3.5%). This included 50 (64.1%) with HCV genotype 1a and 28 (35.9%) with genotype 3a. Fifty-six percent (50 of 90) of genotype 1a participants were involved in clusters, and 58% (28 of 49) of genotype 3a participants were involved in clusters. A total of 12 clusters were identified in the study, including 6 for genotype 1a and 6 for genotype 3a (Fig 3). Among genotype 1a, cluster size ranged from 3 to 15 participants (mean, 8; Q1-Q3: 3–15). Among genotype 3a, cluster size ranged from 3 to 9 participants (mean, 5; Q1-Q3: 3–5, Fig 3). The mean genetic distance in a cluster was 2.2% (Q1-Q3: 1.7% – 2.9%) nucleotide substitutions per site for genotype 1a, and 2.1% (Q1-Q3: 1.3% – 3.0%) nucleotide substitutions per site for genotype 3a. The combined mean genetic distance among non-clustered participants for genotypes 1a and 3a was 3.2% (2.8% – 3.5%).

### 3.3. Clustering patterns and descriptive statistics stratified by phylogenetic clustering

Among participants with an available HCV sequence for genotypes 1a and 3a (n = 141), 78 were members of a cluster. Four clusters are included in this manuscript (Fig 4) whereas clusters with repeated patterns and no clear patterns can be found in S1 Fig. Among the four included clusters, participants clustered with different age groups and gender. A clear distinction in clustering was observed, characterized by an over-representation of participants from a single town or a specific

racial group within each cluster (Fig 4A-D). Participants in cluster 3 (or A) showed distinctive clustering with mostly negative HIV status and use of both methamphetamine and heroin. In cluster 17 (or B), participants were mainly HIV-positive (7 of 9) and used heroin (6 of 9), although 2 did not use either drug.

The descriptive statistics stratified by phylogenetic clustering of factors previously associated with HCV infection and phylogenetic clustering are shown in S2 Table. Factors with a significant difference among those in a cluster (n = 78) and those not in a cluster (n = 63) were age ≥ 29 years (p = 0.045), race (p = 0.000), and city (p = 0.000). The latter were included in the logistic regression model.

### 3.4. Factors associated with phylogenetic clustering

Univariate and multivariate logistic regression analysis of factors associated with phylogenetic clustering are summarised in Table 3. Factors positively associated with HCV phylogenetic clustering on the univariate logistic regression were age ≥ 29 years (OR 2.03, 95% CI 1.01–4.10, p = 0.046), mixed ancestry race (vs. white race) (OR 3.70, 95% CI 1.25–10.97, p = 0.018), and residing in Cape Town (OR 7.14, 95% CI 2.63–19.40, p = 0000). Black race (vs. white race) was negatively associated with phylogenetic clustering in the univariate analysis (OR 0.37, 95% CI 0.16–0.81, p = 0.014).

In multivariate analysis, three factors were positively associated with HCV phylogenetic clustering, including sharing a needle at the last injection (aOR 4.02, 95% CI 1.08–14.87, p = 0.037), age ≥ 29 years (aOR 3.00, 95% CI 1.22–7.37, p = 0.016), and mixed ancestry race (aOR 6.11, 95% CI 1.87–19.95, p = 0.003, Table 3). Sharing a needle at the last injection was associated with clustering when adjusted for age ≥ 29 years and mixed ancestry race. The odds of being in a cluster were 4 times greater if the participant shared an injecting needle, 3 times greater if the age was ≥ 29 years, and

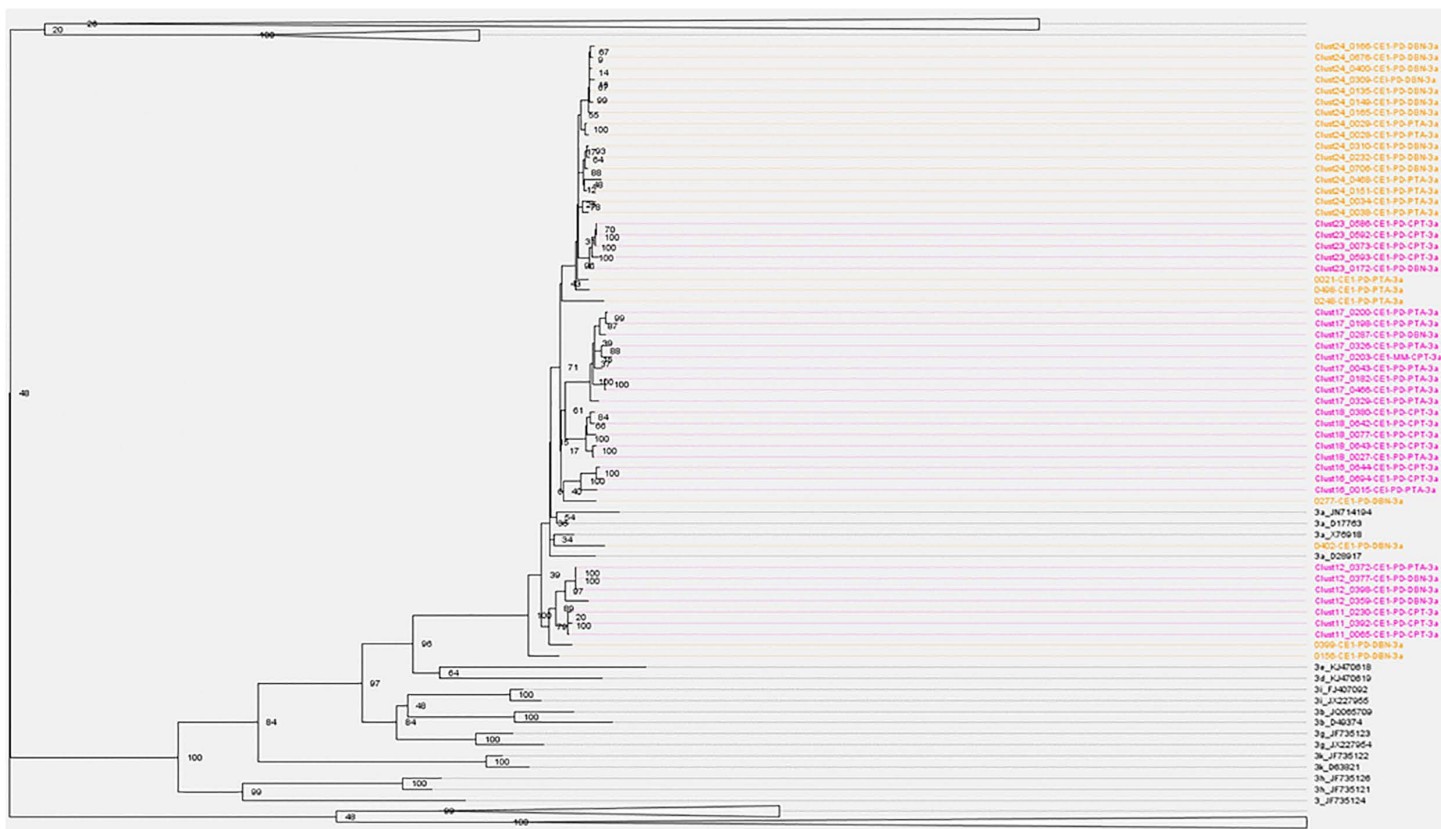

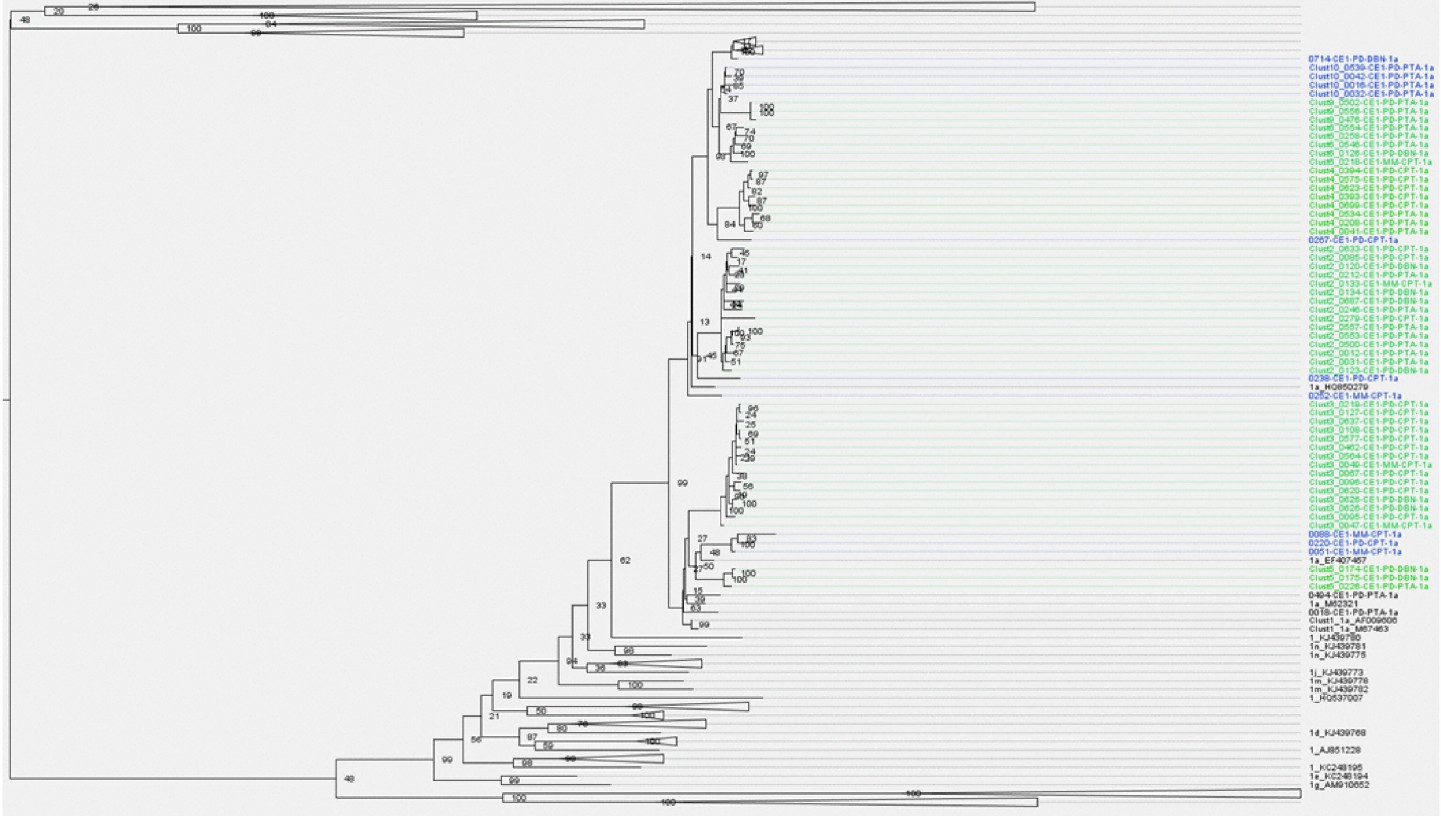

**Fig 2. Phylogenetic tree of HCV South African sequences with Smith panel reference sequences [1].** The maximum likelihood tree was inferred using RAxML, and clusters were identified using ClusterPicker with a bootstrap threshold ≥90% and genetic distance <3.5%. a. Colours differentiate clustered study sequences (genotype 3a = pink) from non-clustered sequences (genotype 3a = orange) and the Smith panel of references (black). b. Colours differentiate clustered study sequences (genotype 1a = green) from non-clustered sequences (genotype 1a = blue) and the Smith panel of references (black). The phylogenetic trees were visualized using Figtree [30].

6 times greater if the participant was of mixed ancestry. Residing in Cape Town was positively associated with clustering in the univariate analysis and not with the adjusted analysis. Further, sharing a needle at the last injection was not associated with clustering in the univariate analysis but was associated with clustering when adjusted for age ≥ 29 years and mixed ancestry race. The quality analysis of the multivariate logistic regression model showed that the model does not lack goodness of fit, and the area under neath the curve (AUC) was 72% (S2 Fig). The median age ≥ 29 years used in this analysis was similar to that of the larger study [11]; the same analysis was conducted using the median age ≥ 31 years found in this study (S3 Table). The same analysis was also conducted exclusively for the PWID sample population to determine whether the same factors were associated to PWIDs and we had seen concurrent results (S4 Table). Additionally, the AUC of from this analysis was 80%.

## 4. Discussion

This study described HCV transmission networks through phylogenetic clustering among PWID in SA. Compared to a prevalence of <1% in the general population [11], our wider study on MSM and PWID showed that the HCV disease burden is concentrated among the key populations in SA, with a seroprevalence of 27% and 55%, respectively [10–12]. The

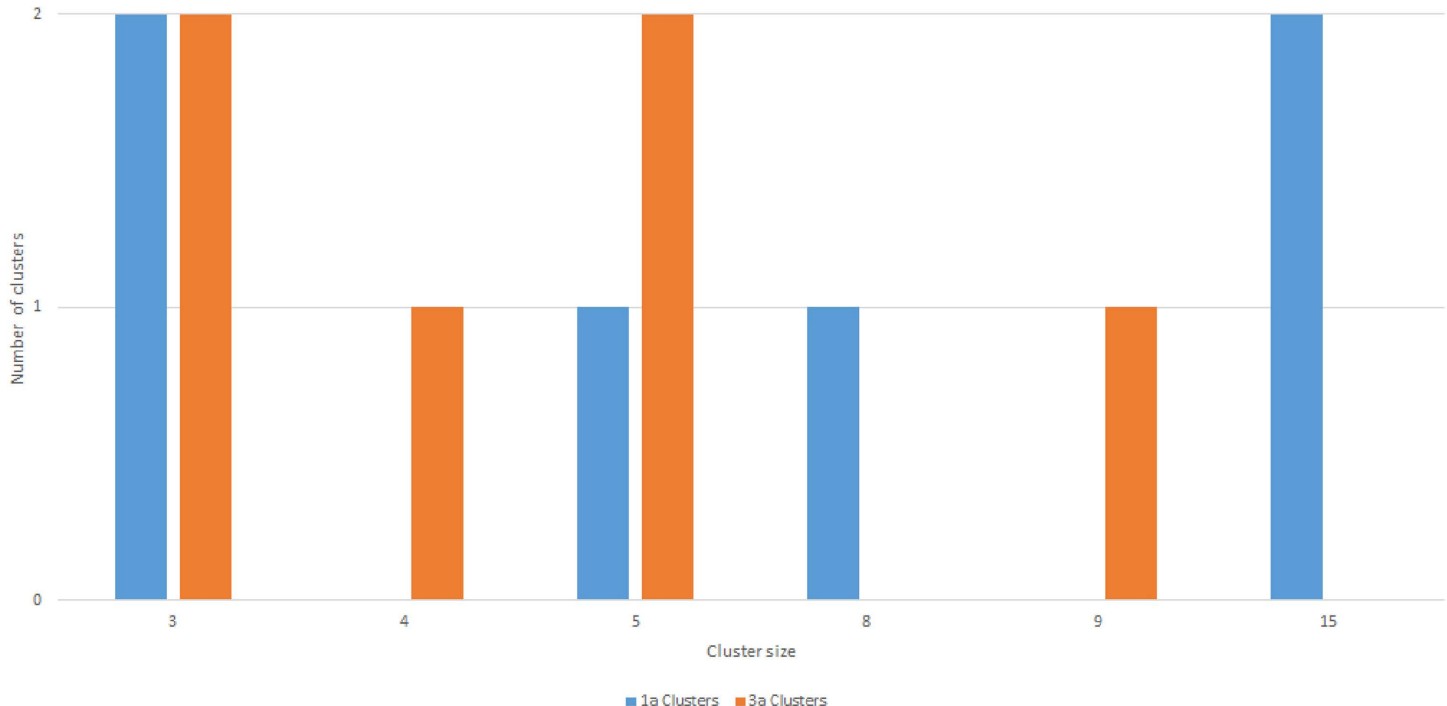

**Fig 3. Genotype 1a and 3a phylogenetic cluster size distribution in three SA cities, 2016–2017.** Participants in clusters (n ≥ 3) differentiated from non-clustered participants using ClusterPicker with a bootstrap threshold of ≥90% and genetic distance of <3.5%.

latter was a cross-sectional study conducted between 2016 and 2017 investigating the prevalence and factors associated with HCV infection among these populations. Investigators did not collect partner tracing information; thus, ongoing HCV transmission networks could not be evaluated.

In this study, an alarming 55% of participants were involved in phylogenetic transmission clusters, higher than previously described in other countries such as Australia and Canada, at 22% and 31%, respectively [13,18]. High clustering suggested that PWID are at increased risk of HCV disease in SA, and there is a high probability of HCV spread among this population. Higher phylogenetic clustering patterns can also be attributed to the study design. In contrast to prospective studies conducted, our study employed a cross-sectional design and recruited participants accessing HIV prevention services. This methodological difference may have contributed to the higher phylogenetic clustering patterns observed in our findings.

The multivariate analysis showed increased odds of phylogenetic clustering among participants that shared a needle at the last injection, and this risk factor was previously reported elsewhere [18]. By contrast to the low numbers of participants whom indicated sharing needles, there seemed to be high transmission clusters. The finding could be a result of favourable reporting because participants were from a lower socioeconomic background, and there have been reports of the inability to afford or access sterile injecting equipment [11,31]. Our results highlight the urgent need to distribute sufficient, high-quality sterile injecting equipment among PWID in SA, ensuring that it aligns with user preferences and the specific substances being abused. Different types of syringes may be required depending on the substances used; for instance, larger bore syringes are often necessary for injecting thicker substances like heroin or stimulants, while smaller bore syringes may be more suitable for opioids [32]. The Global fund recommends a minimum of 300 sterile needles per person per year to effectively reduce HCV transmission and other blood-borne infections [33].

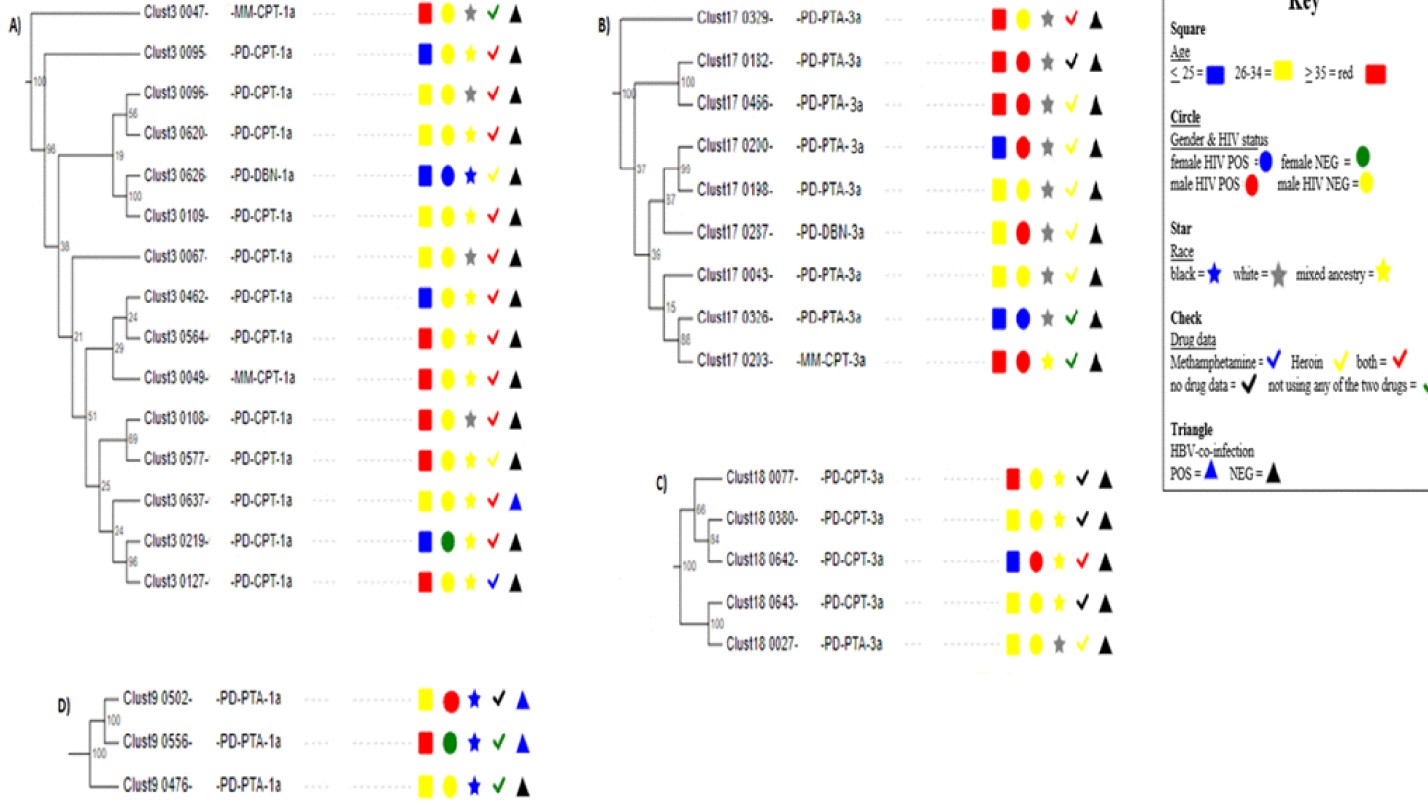

**Fig 4. Maximum-likelihood phylogenetic trees showing clusters with many and few participants for genotype 1a (A &D) and genotype 3a (B &C).** PD represents PWID/UD, MM=men who have sex with men, CPT=Cape Town, DBN=Durban, and PTA=Pretoria. Phylogenetic clusters were identified using ClusterPicker with a bootstrap threshold of ≥90% and a genetic distance of <3.5%. Phylogenetic trees were annotated in Evolview [29].

**Table 3. Univariate and multivariate logistic regression analysis of factors associated with being in a phylogenetic cluster for participants with HCV genotype 1a or 3a.**

| Characteristics | Overall (n=141) | Not in a cluster (n=63) | In a cluster (n=78) | Odds ratio | 95% CI | p | Adjusted odds ratio | 95% CI | p |
|---|---|---|---|---|---|---|---|---|---|
| Shared needle | 19 (13.5) | 5 (7.9) | 14 (17.9) | 2.53 | 0.85-7.56 | 0.096 | 4.02 | 1.08-14.87 | **0.037** |
| New needle | 82 (58.2) | 39 (61.9) | 43 (55.1) | 0.59 | 0.27-1.30 | 0.191 | | | |
| Age ≥ 29 | 91 (64.5) | 35 (55.6) | 56 (71.8) | 2.03 | 1.01-4.10 | **0.046** | 3.00 | 1.22-7.37 | **0.016** |
| **Race** | | | | | | | | | |
| White | | | | 1.00 | | | 1.00 | | |
| Black | 46 (32.6) | 30 (47.6) | 16 (20.5) | 0.37 | 0.16-0.81 | **0.014** | | | |
| Mixed ancestry | 32 (22.7) | 5 (7.9) | 27 (56.9) | 3.70 | 1.25 −10.97 | **0.018** | 6.11 | 1.87-19.95 | **0.003** |
| **City** | | | | | | | | | |
| Durban | | | | 1.00 | | | 1.00 | | |
| Pretoria | 59 (41.8) | 32 (50.8) | 27 (34.6) | 1.43 | 0.61-3.36 | 0.415 | | | |
| Cape Town | 47 (33.3) | 9 (14.3) | 38 (48.7) | 7.14 | 2.63-19.40 | **0.000** | | | |

Additionally, needs-based syringe distribution models have been shown to significantly decrease syringe sharing and improve health outcomes among PWID [34]. Tailoring syringe distribution to the specific needs of PWID can enhance the effectiveness of harm reduction strategies and mitigate health risks associated with diverse injecting practices [35,36]. Initiatives such as the STEP-UP project [31] should be endorsed with more financial support to ensure the WHO target for needles distributed per PWID is met [37]. Further, confiscation of sterile injecting equipment by law enforcement toward PWID [11,31] should be stopped. Communities must be continually informed about substance use disorders and the necessary means to access treatment and harm reduction interventions. It is crucial to recognize that mere sensitization has not been shown to effectively reduce HCV incidence among PWID [38]. Therefore, there is an urgent need to rapidly upscale treatment and harm reduction interventions, including screening and simultaneously treating networks of PWID.

The multivariate logistic regression also identified increased odds of clustering for mixed ancestry race. This finding is novel as it was not described in the larger study and provides insight into the transmission dynamics by race in SA. Results suggest that transmission occurs among people who are acquainted with each other. From phylogenetic clusters, most participants clustered locally, with similar races, HIV status, and drug use patterns. A tailored non-share needle education could be provided for clustered participants with their networks [39]. There was a notably concentrated methamphetamine use among the mixed ancestry race, which is also visible in a phylogenetic cluster. Methamphetamine use may be a potential contributing factor that driver infection in Cape Town among those of mixed ancestry race. This finding has been previously described among a group of PWID elsewhere [20,40]. Further, there has been an increase in methamphetamine use as a primary drug in Cape Town over the years, and drug prices have also been relatively stable [41].

The associations observed in this analysis suggest potential shared transmission dynamics among HIV, HBV, and HCV, as indicated by the sequence similarity of viral strains within phylogenetic clusters. Previous studies have documented shared transmission routes for these viruses, highlighting how genetic relationships can reflect common pathways of infection [42]. Overall, 30% (43/141) of the study participants were HIV positive, but there was no significant difference in the number of HIV positive among clustered and non-clustered participants. In contrast to the previous study, HIV was associated with HCV infection (or acquisition) [11]. HCV and HIV prevalence was the highest in Pretoria, potentially related to the long-existing drug injection in the area compared to the other cities [11]. We did observe low specificity in the amplification and sequencing assay in the current study, which may be related to older circulating lineages in the Pretoria area. Very few cases of HBV infection were identified in the analysis and represented by phylogenetic clustering. However, the prevalence of HBV in this population appears to align with that of the general population [12,43,44], suggesting that many cases may be childhood-acquired rather than linked to recent injection practices.

This study had several limitations. Our study was a cross-sectional study between 2016–2017, and participants were recruited through harm reduction services. Therefore, the findings cannot be generalized to the broader population of SA. The phylogenetic clustering of participants from different geographical locations or cities was challenging to interpret. Our study mainly focused on key populations from lower socioeconomic status and mostly the homeless. It is possible that they were infected in one city and then migrated to another, but also less likely in view of their socio-economic status. Similar scenarios have been reported in other settings and interpreted as missing linkages in the transmission network [13,18]. Additional participants who were part of the clustering network may have been missed due to sequencing fails and certain participants not being recruited in the study.

Information on each participant's infection date was not collected and is unknown. Additionally, establishing the precise age of infection in study participants can be challenging or may not be accurate when self-reported. The results of this analysis could be influenced by the timing of infection, even though we did not specifically consider this factor. Genetic distances among viral strains are typically shorter for recent infections compared to those that have been established for a longer duration. This is due to the error-prone nature of the key enzyme involved in HCV replication, which leads to increased genetic variation over time [45].

Additionally, the direction of HCV transmission, which refers to the pathways through which the virus spreads between individuals and specifically identifies who infected whom within a network, was not assessed and therefore cannot be determined. Further, some samples from Pretoria failed the complete sequencing of the Core-E2, which may reflect virus

evolution. Thus, modified primers may be required to overcome these failures. Caution should be exercised when looking at HIV prevalence for the small number of samples sequenced for MSM and PWUD. Additionally, the small sample size led to fewer variables (e.g., MSM and PWUD) that could be adjusted on the logistic regression model and produced wide confidence intervals. Lastly, inherent limitations from the larger study could not be eliminated. The sampling method limits the generalizability of findings to larger PWID populations, despite alignment with previous studies. Information bias may have led to underreporting of substance use and sexual activity, particularly anal sex and sex for drugs. Low reports of substance use during sex in Durban likely underestimate actual behaviors, possibly due to questioning approaches or social norms. While researchers' familiarity may have made participants feel safe, it could also result in underreporting risky behaviors and overestimating access to needle and syringe services, skewing HCV risk perceptions.

## 5. Conclusion

The high phylogenetic clustering underscores the concentrated burden of HCV among PWID in SA, particularly among mixed ancestry individuals in Cape Town, and the findings highlight the urgent need for targeted interventions. To reduce HCV transmission, it is crucial to provide safe injecting equipment and implement tailored educational programs aimed at preventing transmission between older and younger drug users. The findings indicate that needle sharing significantly increases the risk of clustering, necessitating the distribution of at least 300 sterile needles per person annually, as recommended by the Global Fund. Additionally, addressing the specific needs of PWID regarding syringe types based on substance use is essential for effective harm reduction strategies.

Despite these contributions, significant research gaps remain. The cross-sectional design limits generalizability and the lack of data on infection timing constrains our understanding of transmission dynamics. Future studies should adopt longitudinal designs to explore these dynamics and assess intervention effectiveness. In summary, there is an urgent need to rapidly upscale harm reduction efforts and enhance education about substance use disorders to mitigate HCV transmission within PWID. Continued research is essential to address existing gaps and inform effective public health strategies aimed at reducing HCV spread in South Africa.

## Supporting information

**S1 Table. Substance use and sex practice of participants with Core-E2 sequences.**
(DOCX)

**S2 Table. Characteristics of participants with Core-E2 sequence in a cluster, not in a cluster, and overall.** Participants with HCV genotype 1a or 3a were included in the analysis.
(DOCX)

**S3 Table. Univariate and multivariate logistic regression analysis of factors associated with being in a phylogenetic cluster for participants with HCV genotype 1a or 3a.**
(DOCX)

**S4 Table. Univariate and multivariate logistic regression analysis of factors associated with being in a phylogenetic cluster for participants with HCV genotype 1a or 3a.** Analysis exclusively for PWID.
(DOCX)

**S1 Fig. Maximum-likelihood trees showing clusters with many and few participants for genotype 1a (A-D) and genotypes 3a (E-H).**
(TIF)

**S2 Fig. Area under ROC curve and goodness of fit test for the multivariate logistic regression model.**
(TIF)

# Acknowledgments

Special appreciation is extended to the participants of the study. We also wish to recognize the contributions and involvement of the following individuals: Lorraine Moses, Andrea Schneider, Kalvanya Padayachee, Dawie Nel, Nelson Medeiros, Angela McBride, Kevin Rebe, Ruth Motlafi, Penelope Daki, Yolaan Andrews, Andrew Lambert, Rudolph Basson, Wendy Joyce, Joezette MacKay, Angelina Satira, Mfezi Mcingana, Lesedi Mohlalal, Zara von Hofmeyer, Laurene Booyens, Jayendrie Thaver, Lucinda Gaelejwe, Jack Manamela, Susan Malfeld, Lillian Makhathini, Angella Musisi, and the teams responsible for implementing the study and supporting laboratory procedures. The implementation at sites managed by TB HIV Care was in addition to the HIV services provided by the United States President's Emergency Plan for AIDS Relief (PEPFAR), the Centers for Disease Control and Prevention (CDC), and the Global Fund. The National Institute for Communicable Diseases, Centre for Vaccines and Immunology, for their contributions to genomic sequencing testing and related data analysis.

# Author contributions

**Conceptualization:** Nkosenhle L. Ndlovu, Nishi Prabdial-Sing.

**Data curation:** Nkosenhle L. Ndlovu, Andrew Scheibe.

**Formal analysis:** Nkosenhle L. Ndlovu.

**Funding acquisition:** Nkosenhle L. Ndlovu, Andrew Scheibe, Nishi Prabdial-Sing.

**Investigation:** Nkosenhle L. Ndlovu, Andrew Scheibe, Mark W. Sonderup, C. Wendy Spearman, Jason T. Blackard, Nishi Prabdial-Sing.

**Methodology:** Nkosenhle L. Ndlovu, Andrew Scheibe, Jason T. Blackard, Nishi Prabdial-Sing.

**Project administration:** Nkosenhle L. Ndlovu.

**Resources:** Nishi Prabdial-Sing.

**Software:** Nkosenhle L. Ndlovu, Jason T. Blackard.

**Supervision:** Jason T. Blackard, Nishi Prabdial-Sing.

**Validation:** Nkosenhle L. Ndlovu.

**Visualization:** Nkosenhle L. Ndlovu.

**Writing – original draft:** Nkosenhle L. Ndlovu.

**Writing – review & editing:** Nkosenhle L. Ndlovu, Andrew Scheibe, Harry Hausler, Mark W. Sonderup, C. Wendy Spearman, Katherine Young, Dawie Nel, Jason T. Blackard, Nishi Prabdial-Sing.

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
