## [Decision Letter · Decision Letter 0]

30 Jun 2025

Dear Dr. Ndlovu,

Thank you for submitting your manuscript to PLOS ONE. After careful consideration, we feel that it has merit but does not fully meet PLOS ONE’s publication criteria as it currently stands. Therefore, we invite you to submit a revised version of the manuscript that addresses the points raised during the review process.

Specifically, the manuscript lacks clarification on certain points in the methods, results, and discussion. Clarifying those points, as outlined in the reviewer comments, can enhance the quality and readability of the manuscript.

We look forward to receiving your revised manuscript.

Kind regards,

Syed Hani Abidi

Academic Editor

PLOS ONE

Journal Requirements:

Reviewers' comments:

Reviewer's Responses to Questions

**Comments to the Author**

1. Is the manuscript technically sound, and do the data support the conclusions?

Reviewer #1: Yes

Reviewer #2: Yes

2. Has the statistical analysis been performed appropriately and rigorously?

Reviewer #1: Yes

Reviewer #2: Yes

3. Have the authors made all data underlying the findings in their manuscript fully available?

Reviewer #1: Yes

Reviewer #2: Yes

4. Is the manuscript presented in an intelligible fashion and written in standard English?

Reviewer #1: Yes

Reviewer #2: Yes

Reviewer #1: Nkosenhle Lindo Ndlovu, Andrew Scheibe and co- authors have conducted a molecular epidemiological study titled “Factor associated with phylogenetic clustering of hepatitis C virus, mainly among people who inject drugs who access HIV prevention services in South Africa , 2016-2017. They included 285 samples for genotyping, of which 141 were sequenced and used to study genetic and demographic information for phylogenetic clustering analysis. This is a well written and timely study addressing a critical public health issue. The objective of the study is clear, the introduction is informative, the methodology technically sounds, the data are well analyzed, and results are presented clearly. The study findings provide valuable insights into hepatitis C virus transmission dynamics among PWID in South Africa.

Overall, the manuscript is in good shape and suitable for publication in PLOS One. I have two comments that should be addressed to improve the clarity and presentation.

Comments:

1. Page 14, line 116: it is said that samples were collected from PWID, MSM and PWUD populations. However, in the abstract, only PWID and MSM are mentioned as the source of sequences used for phylogenetic clustering. The PWUD inclusion is not explained in the abstract. PWUD stands for?

2. In the tree of Fig 2. The authors used Smith Panel sequences as reference sequences for phylogenetic analysis. However, the tree appears noisy and difficult to interpret. It would be helpful if the authors could improve the visualization. One possible approach to improve clarity would be to initially use all reference sequences for classification. Once the sequences of interest are grouped with relevant reference clades, a refined tree could be constructed using only the most relevant subset of reference sequences. This would enhance the readability of the final tree while maintaining the accuracy of the phylogenetic relationships.

Reviewer #2: PONE-D-25-15243

Research Article Title: Factors Associated with Phylogenetic Clustering of Hepatitis C Virus, Primarily Among People Who Inject Drugs Accessing HIV Prevention Services in South Africa, 2016-2017.

Author: Nkosenhle Lindokuhle Ndlovu, M.D.

Thank you for the opportunity to review this research article, which addresses a significant public health problem in South Africa. This study received approval from the University of the Witwatersrand Human Research Ethics Committee (clearance certificate M181169), a copy of which was included in the documents shared for review.

The sample and study data were drawn from a previous study that examined viral infections in people who inject drugs (PWID) and men who have sex with men (MSM) in South Africa (with references to publications from this project shared).

Hepatitis C virus (HCV) predominates in key population groups, with genotypes 1a and 3a being the most common identified in this study. This study effectively uses molecular epidemiology to genetically link related cases across various South African cities.

The study is well written, with clear objectives, study methodology and study findings/discussion.

Comments, Clarifications, or Queries:

1. Were there participants with mixed infections, for example, with both 1a and 3a genotypes, possibly due to cross-infections?

2. The shared letter of approval indicates that the study was approved for four cities in South Africa, yet the study itself mentions only three. Please clarify this discrepancy.

3. Regarding Table 1A, what does "N/A" signify when referring to the MSM population for Pretoria and Durban? Does it indicate missing data or no MSM populations recruited into the study?

4. What was the study design for the original study from which the samples for this research were obtained? Additionally, what was the study design for samples acquired from the "specialist clinic in Cape Town"?

5. This study has several acknowledged limitations. It is also worth noting that establishing the precise age of infection in study participants can be challenging or may not be accurate when self-reported.

**Do you want your identity to be public for this peer review?** For information about this choice, including consent withdrawal, please see our Privacy Policy

Reviewer #1: No

Reviewer #2: No

---

## [Author Response · Author response to Decision Letter 1]

1 Oct 2025

Thank you for the valuable insights and supporting our work!

---

## [Editor Report · Decision Letter 1]

28 Oct 2025

Factors associated with phylogenetic clustering of hepatitis C virus, mainly among people who inject drugs who access HIV prevention services in South Africa, 2016-2017.

PONE-D-25-15243R1

Dear Dr. Ndlovu,

We’re pleased to inform you that your manuscript has been judged scientifically suitable for publication and will be formally accepted for publication once it meets all outstanding technical requirements.

Kind regards,

Syed Hani Abidi

Academic Editor

PLOS ONE
---

## [Editor Report · Acceptance letter]

PONE-D-25-15243R1

PLOS ONE

Dear Dr. Ndlovu,

I'm pleased to inform you that your manuscript has been deemed suitable for publication in PLOS ONE. Congratulations! Your manuscript is now being handed over to our production team.

Kind regards,

on behalf of

Dr. Syed Hani Abidi

Academic Editor

PLOS ONE